# Environmental Adaptation in the Process of Human-Land Relationship in Southeast China’s Ethnic Minority Areas and Its Significance on Sustainable Development

**DOI:** 10.3390/ijerph20032737

**Published:** 2023-02-03

**Authors:** Zhi Zhang, Zhiwei Wan, Shan Xu, Hong Wu, Lingyue Liu, Zhao Chen, Ji Zeng

**Affiliations:** 1School of Ecology and Environment, Yuzhang Normal University, Nanchang 330103, China; 2School of Geography and Environmental Engineering, Gannan Normal University, Ganzhou 341000, China; 3Development Research Center of the State Council, Beijing 100010, China; 4National-Regional Joint Engineering Research Center for Soil Pollution Control and Remediation in South China, Guangdong Key Laboratory of Integrated Agro-Environmental Pollution Control and Management, Institute of Eco-Environmental and Soil Sciences, Guangdong Academy of Sciences, Guangzhou 510650, China

**Keywords:** environmental adaptation, GIS spatial analysis, place name cultural landscape, southeastern China, She nationality

## Abstract

The relationship between regional human development and geographic environment is the basis for dynamic social change, and studying the evolution of human-land relations in typical regions can provide background knowledge for global change studies. This study is based on GIS and spatio-temporal statistical techniques, combined with the analysis of toponymic cultural landscapes, to study ethnic minority regions of southeastern China. The results show that: (1) The geographical environment of the region will affect the naming of villages, and the orientation and family name are the most common; the frequency of plants, pit (*keng*), animals, and flat (*ping*) is also very high. (2) Han settlements and She settlements have obvious spatial differentiation, and in general the Han distribution area is lower than that of the She. Han settlements are mainly distributed in plain areas along rivers with elevations less than 200 m; She settlements are mainly distributed in hilly areas (200~500 m) and low mountain areas (500~800 m). (3) The results of quadrat analysis and nearest neighbor index analysis show that both Han and She settlements are clustered in the spatial distribution pattern, and the distribution of She settlements is more clustered than that of Han, with more dense settlements at a certain spatial scale. The regional cultural landscape is the result of the development and evolution of human-land relationship, and the comprehensive analysis of cultural landscape can understand the process of human-land relationship in a small region. The settlements in the region are indicative of the geographic environment in terms of village naming, spatial pattern, elevation differentiation and relationship with rivers, which can reflect the environmental adaptation process of human activities.

## 1. Introduction

The Human-Land relationship is the abbreviation of dynamic relationship between human society and the natural environment [1], in which human beings and nature are closely linked, interact and interdepend on each other. On the one hand, nature provides human beings with survival conditions, meets the basic needs of human beings, and greatly enriches human life [2]; human activities will in turn affect nature and even transform environment [3]. Therefore, man-land relations are of great significance in scientific research [4]. The development of human society is based on the interaction between human and nature relations. Human-land relations are not only the core areas of geography research, but also have gradually developed into an important part of the global changes in scientific research programs [5]. The spatial distribution of Tibetan villages on the Qinghai-Tibet Plateau in northwest China [6] and the spatial analysis of Dai and other ethnic villages in southwest China [7] show that the geographical environment is an important factor affecting the distribution of settlements.

The development of human-land relationships is a long-term scale development process, but due to the lack of data and monitoring methods, most of China’s research periods are more concentrated after 1949, especially after China’s reform and opening up (1978~). The southeast of China is the main gathering area of the population and one of the most social and economic development areas in China [8]. Therefore, this area is the most dense and concentrated area for the interaction between human-land relationship in China, and has certain representativeness and instructions in global development. Studying the evolution of human-land relations during the historical period can be used as an important regional case for the evolution of global changes [9].

Cultural landscape is an important part of human-land relationships, and one of the main directions of national science, sociology, and human geography [10,11]. It is generally believed that “landscape” is a geographical complex, which is generated by nature and humanistic phenomena [12]. Therefore, as the carrier of human-land relationships, “place name” is the synthesis of human phenomena in specific regions [13]. It can in-depth discussions of the dynamic pattern of social phenomena through the research of geographical name cultural landscapes [14]. In recent years, with the widespread application of Geographic Information System (GIS) in social sciences [15], many scholars have begun to use GIS spatial analysis methods to study place names and cultural landscapes [16]. With the further deepening of research, the study of the national regional place names and cultural landscapes at a smaller scale is particularly important. Especially in the traditional Han people’s life areas in eastern China, some ethnic minority areas that are mixed with the Han people have different cultural landscape characteristics. As the Han nationality in this area entered the process of civilization earlier, the development of its human relations was more affected by other advanced culture and technology. Therefore, in order to better understand the characteristics of the evolution of human-land relationship before industrialization, we can try to study the relatively undeveloped regions where less affected by human activities.

Although human-land relations can be reflected in multiple aspects, the process of this relationships in the historical period is difficult to be preserved at a long time, so it is hard to study it. The place name is a comprehensive product of human interaction with the environment, and it is a cultural specific form of man-land relationships. In recent years, many scholars have started research on historic and local relations with geographical cultural landscapes in ethnic regions. *Chinese Encyclopedia* [17] defines geographical names as a proprietary name of natural or human geographic entities in a certain space position. *The Spring and Autumn Annals* [18] records the view of “water north is *yang*, and the south of the mountains is *yang*”. This also shows that people will name the settlement by summarizing a certain geographical environment. According to the analysis of a large number of ethnic place names, research shows that the place name can reflect the degree of settlement and dissemination of ethnic groups in a specific area, and to a certain extent reflect the dynamic evolution of the spatial distribution of the ethnic composition to a certain extent. It can be seen that the study of place names is a subject that is closely related to historical geography, archeology, ethnology, sociology, etc. [19,20]. Through the research of place names, we can provide a new explanation and perspective for the research of related social sciences. For example, Wang et al. [21] using spatial interpolation technology to obtain the spatial distribution trend of Zhuang nationality language in Guangxi, and reproduce the distribution and migration process of Zhuang people in history.

Chinese ethnic minorities can generally be divided into two main forms: concentrated residence and mixed residence. The number of ethnic minorities in the mixed residence region accounts for about one-third of the total population of ethnic minorities. In the past, most of the research discussed the form of ethnic dwellings in large areas [22,23]. Since this study mainly focuses on the distribution of ethnic settlements on a small scale (for example, towns), the impact of the environment on settlements on a small scale is more obvious than that on a large scale (for example, provinces or cities). For example, all ethnic groups are mixed with Yunnan Province, which are more common in Yunnan Province, and the Han people are distributed in plains and eastern regions. The Dai people are mainly living in the river valley; Hani, Jingpo and other ethnic groups live in half mountains. The Nu, Dulong etc. are mainly concentrated in the mountains [24]. Some research also focused on the mixed living conditions of ethnic minorities and other nationality in the urban community [25]. However, this phenomenon is mainly due to special historical processes and the result of urbanization, and it does not reflect the dynamic process and mechanism of interaction between man-land relations interaction in ethnic minority areas in history.

In recent years, with the development of spatial statistics technology [26,27], relevant scholars have begun to use quantitative analysis methods to study the evolution process of man-land relations at different scale [28,29]. Further, study the relationship between the spatial distribution of ethnic minorities and the geographical environment. Overall, there are few researches on the study of national and cultural landscapes in small scale, especially geographical environmental factors on national development mechanisms in the same area [6,7]. As an important part of the 56 ethnic groups in China, the She nationality is mainly distributed in the mountainous areas of Jiangxi, Fujian, and Zhejiang provinces in southeast China.

The northeastern area of the Jiangxi Province is the most concentrated area of the She people in Jiangxi Province. Among them, the southern mountainous areas of Yingtan City and Shangrao City are in the core area of the She ethnic groups, which are representative. There are few studies on the spatial distribution of villages in this area and its influencing factors. And there are few studies on the evolution and interaction between human and environment in this region. According to the results of studies in other regions, we speculate that the spatial distribution pattern of different ethnic settlements in this region may be affected by the natural environment and production mode. Therefore, this study selects Zhanping Township in Yingtan City, Taiyuan Township and Tianzhushan area in Shangrao City as the study area, and uses GIS and spatial statistical techniques to analyze the differences between She and Han cultural landscapes and the evolutionary characteristics of man-land relations in this area. The results of the study can provide explanations and motives on environmental elements to further explore the historical interaction process and the evolution of human-land relations between the She and Han peoples.

She nationality is one of the major minority groups in southeast China, mainly scattered in the border areas of Fujian, Zhejiang, Jiangxi and Guangdong. The total population of She nationality is about 746,000, accounting for 0.05% of the total population of the country [30]. The She people do not have written language commonly used by the local ethnic groups, but have their own oral language, which belongs to the Miao and Yao language group. The main form of livelihood of the She people is “slash-and-burn farming” [30]. Jiangxi has been the main settlement of the She ethnic group, currently there are 7 She townships, in addition to 77 She administrative villages scattered. In this study, Zhanping Township, Taiyuan Township and Tianzhushan areas, located in the core area of She, at the northern foot of Wuyi Mountain, were selected as the study area, where Zhanping and Taiyuan are the first two She townships established in Jiangxi Province(1954~), for the reason that the She population in this region is the most concentrated. Since 1949, the administrative division of the Tianzhushan area has changed frequently, with the establishment of forestry and reclamation farms and the inclusion of towns such as Huangbi, which are geographically and spatially connected and belong to the same She cultural area.

## 2. Overview of the Study Area

The geographical range of the study area is between 117°18′50″ E and 117°43′23″ E; and between 27°50′01″ N and 28°02′59″ N (Figure 1). The area runs roughly west-east, with a north-south length of about 30 km and an east-west width of about 42 km. This area belongs to the central subtropical warm and humid mountain climate, the average annual temperature is 15 °C, the annual rainfall is 1700 mm, the relative humidity is 80%, the soil is mainly red loam, both yellow loam and yellow-brown loam, the vegetation is typical of the central subtropical evergreen broad-leaved forest, the forest coverage rate is above 90% [31]. The region has less arable land and relies mainly on nomadic farming and hunting to sustain its livelihood, with arable land per capita below 400 m^2^.

## 3. Data and Methods

### 3.1. Data Sources

The data of the place names in the study area were obtained from the Atlas of Jiangxi Province and Jiangxi gazetteers, with a total of 232 village place names [32,33,34]. The basic data such as boundary and river data were obtained from the national 1:250,000 basic geographic information database, and georeferenced in ArcGIS 10.2 software according to the administrative map of Jiangxi Province [35]. The Digital Elevation Model (DEM) data were obtained from the SRTM1 DEM dataset published by the National Aeronautics and Space Administration (NASA) (https://earthexplorer.usgs.gov/, accessed on 27 September 2022).

### 3.2. Research Methodology

#### 3.2.1. Research Framework

Firstly, based on ArcGIS software platform, all maps, name place point, basic geographic data and DEM data within the study area are spatially georeferenced, and geodatabase is established to realize accurate matching of spatial information and name place attribute information. Second, the Spatial Statistics Tools toolbox in ArcGIS was used to perform spatial statistics on all spatial attributes, including place names, elevation, etc. Then, the thematic map function in ArcGIS Layout mode is used for spatially differentiated mapping of different types of place names. Using Reclass tool, based on the 30 m resolution DEM topographic data, the distribution ranges of different geomorphic type areas were extracted according to the 200 m, 500 m, 800 m and 1200 m contours according to the elevation distribution status, and the distribution pattern characteristics of She and Han settlements in different geomorphic units were statistically analyzed by overlaying with the settlements. The buffer generation tool was then used to generate ranges of 200 m, 400 m, 600 m and 800 m from the water source based on the river. Overlay analysis was performed with the settlements to count the number of She and Han settlements at different distance ranges from the river. Finally, the Spatial Analyst Tools toolbox was used to quantify different place names and topography to further analyze the geo-environmental significance of the evolutionary process of regional human-land relations.

#### 3.2.2. Quadrat Analysis

Quadrat Analysis (*QA*) is the basis of point pattern analysis, a method that counts the number of points in each grid by covering the study area with a set of square grids [36]. The method measures the aggregation or dispersion status of the point pattern by the metric of the ratio of the standard deviation to the mean, which is calculated as,
(1)$$QA=s\;/\;\bar{x}$$
where, $s=\sqrt{\frac{1}{n-1}\sum {\left(xi-\bar{x}\right)}^{2}}$, $\bar{x}=\frac{1}{n}\sum xi$, *x_i_* is the number of settlements in the *i*-th grid. When the value of *QA* = 1, it means that the point pattern is randomly distributed; if *QA* > 1, the point pattern is aggregated; if *QA* < 1, the point pattern is dispersed.

#### 3.2.3. Nearest Neighbor Index

The Nearest Neighbor Index determines whether the distribution of points in space appears to be clustered by comparing the observed value of the average distance of the nearest neighboring point pairs with the expected value of the average distance in a random state [37], calculated as follows:(2)NNI=d(NN)/d(ran)
where, $d(NN)=\sum_{i}^{n}\min(d_{ij})/n$, $d(ran)=0.5\sqrt{A/n}$, A is the total area of the study area and *n* is the number of settlements. *NNI* > 1 means that the spatial point pattern is discrete, *NNI* < 1 means that the spatial point pattern is aggregated, and when *NNI* = 1, it means that it is randomly distributed.

#### 3.2.4. Ripley’s K Function

In order to explore the change trend of point pattern at different spatial scales, Ripley proposed to use the K function to measure the change of point pattern with distance, so the method can analyze the characteristics of point pattern at arbitrary scales. Ripley’s K(d) represents the ratio of the number of points within the radius of the observation range with d and the density of points in the region [38,39]. It is calculated as follows,
(3)$$K(d)=A\sum_{i}^{n}\sum_{j}^{n}w_{i}{}_{j}(d)/n^{2}$$
where, A is the total area of the study area, *n* is the number of settlements, and $w_{i}{}_{j}(d)$ is the number of points within a distance d. In order to maintain the stability of the variance of K(d), the general treatment is to subject it to an square transformation, $L(d)=\sqrt{K(d)/\pi }$. When L(d)-d > 0, it means that the spatial point pattern is aggregated; when L(d)-d < 0, it means that the spatial point pattern is discrete. The value of the L(d) function is the intensity of aggregation, and the larger its value, the higher the degree of aggregation. If L(d) exceeds the confidence interval generated by Monte Carlo simulation, it indicates that the degree of aggregation or dispersion is significant.

#### 3.2.5. Standard Deviational Ellipse

Standard Deviational Ellipse (SDE) is often used to analyze the direction, centrality and overall characteristics of the distribution of points in space [40,41]. The long axis of the ellipse represents the maximum diffusion direction of the overall spatial distribution, the short axis represents the minimum diffusion direction, and the center of the ellipse represents the centrality of the overall distribution of each sample point. The rotation angle (θ) is expressed as the angle between the long axis of the ellipse and the due north direction, indicating the direction of the spatial distribution of the sample points. The size of the standard deviation ellipse area indicates the dispersion of the distribution of sample points, and the smaller the area, the more concentrated the distribution is near the center.

#### 3.2.6. Kernel Density Estimation

Kernel Density Estimation (KDE) is a nonparametric test that can be used to perform analysis of the density of spatial point distribution [42,43]. The basic principle is to estimate the theoretical distribution of sample points in a region by means of a kernel density function, and to convert the discrete sample point density into a density value that is continuously distributed in space [44]. The kernel density analysis can identify the concentrated areas of spatial point element distribution, which are also known as hot spot distribution areas. The calculation is as follows,
(4)$$F_{n}(x)=\frac{1}{nr}\sum_{i=1}^{n}k(\frac{x-x_{i}}{r})$$
where, *k*(·) is the kernel function, *r* is the analysis radius, and $x-x_{i}$ is the distance between the point *x* to be estimated and the sample point *x_i_*.

## 4. Results

### 4.1. Distribution Characteristics of Toponyms

#### 4.1.1. Word Frequency Distribution Characteristics

There are 232 toponymic points of various types in the study area, and the naming elements can be divided into the following categories. First, named after topographical features, such as *Ping* (flat), *Keng* (pit), *Yan* (rock), *Bei* (back), etc.; second, named after family names, such as *Hejia* (He Family), *Niejiazhuang* (Nie Family), etc.; third, named after plants and animals, such as Fox Rock, *Huangmagang* (Yellow Horse Hillock), etc.; fourth, named after other features, such as *Xinwuli* (Inside the New House), etc. Word frequency statistics can extract 20 main elements from place names (Figure 2), among which orientation (n = 60) and family name (n = 67) are the most common; plants (n = 23), *keng* (pit, n = 24), animals (n = 17) and *ping* (flat, n = 13) also appear very frequently; in addition, such as *gang* (hillock), *fan* (farmland), *tian* (paddy field), *ao* (depression), *pong* (clump), *tan* (beach) and *ling* (ridge) also appear with certain frequency.

#### 4.1.2. Spatial Distribution Pattern

Different types of toponymic points are spatially displayed on the basic geographic base map (Figure 3), and the characteristics of different toponyms in terms of spatial distribution can be observed. Figure 3 shows the spatial distribution of the toponymic points named by the elements of orientation, family name, plant. From the Figure 3, it can be seen that the distribution of different types of toponymic points has certain divergent characteristics in spatial distribution. The distribution of “orientation” and “family name” is generally consistent with the overall distribution trend of toponymic points in the whole study area, which indicates that these two naming methods have certain universality in the area and belong to the intuitive experience of human activities. The expansion of human activities generally radiates from one point to the periphery, so it is logical to name a new place name by the orientation relationship with the old one. On the other hand, the family name is the symbol of a community, and the new place name is likely to use the name of the developer. The place names named with the element of “plant” are mainly located in the green areas of plains and low altitudes in Figure 3c, which is probably related to the fact that these places are more suitable for agricultural production and human activities, and the place names are more often named with the names of various crops and cash crops, such as “*Zhuye wu*” probably related to the fact that these places are more suitable for agricultural production and human activities. The names of “*keng* (pit)”, “*shan* (mountain)” and “*ping* (flat)” are directly related to the local geomorphology. For example, “*keng* (pit)” generally refers to a concave place, while “*ping* (flat)” refers to a local flat area in mountainous and hilly areas.

### 4.2. Spatial Pattern Analysis

#### 4.2.1. Results of Quadrat Square Analysis

Generate a 2 km × 2 km grid covering the study area (Figure 4), and the mean and variance of the number of settlements in each grid were calculated, and the QA index was finally obtained. The results showed that the QA = 1.022 for all 232 settlements in the study area, indicating that in general, the settlements in the study area are close to the state of uniform distribution. The QA = 1.267 for Han nationality and QA = 1.956 for She nationality residents; this indicates that both Han and She residents are clustered in the spatial distribution pattern, and the distribution of She residents is more clustered than Han. It can also be seen from Figure 4 that the distribution of Han settlements in the study area is larger and more dispersed, while the distribution of She settlements is relatively concentrated, mainly in the central and eastern mountainous areas.

#### 4.2.2. Results of Nearest Neighbor Index

The Average Nearest Neighbor tool was used to calculate the NNI index for the settlements. The results show that the NNI of Han settlements = 0.693 (observed mean distance of 721 m) and the NNI of She settlements = 0.758 (observed mean distance of 931 m). The NNI of both She settlements and Han settlements were less than 1, indicating that both were clustered and distributed, but the distance values between She settlements were smaller than those of Han, indicating that She settlements showed a more concentrated state and the distance between settlements was closer.

#### 4.2.3. Scale Characteristics Reflected by Ripley’s K

Ripley’s K-function and its transformed L(d) function were calculated separately for She and Han by the Multi-Distance Spatial Cluster Analysis tool. The results are shown in Figure 5. The values of L(d) function of She settlements are over the diagonal in the range of <7000 m, that is, She settlements are aggregated in this scale range. The maximum aggregation radius of She settlements is 3510 m. The L(d) function values of Han settlements are clustered within <8500 m, and the maximum clustering radius is 4625 m. The L(d) function values of both She and Han settlements exceed the confidence interval generated by Monte Carlo simulation in the maximum clustering radius, and are significantly clustered. The results of Ripley’s K analysis further validate the results of the sample square analysis and the results of the nearest neighbor index analysis; the aggregation range of Han settlements is larger than that of She settlements, indicating that the distribution area of Han settlements is larger than that of She settlements.

### 4.3. Standard Deviation Ellipse Analysis Results

The results of the analysis of the standard deviation ellipse show (Figure 6) that the center of all settlements (117.53° E, 27.97° N) is distributed in the border area of Zhanping Township and Taiyuan Township. The She center is located about 4500 m northwest of the center point, and the Han center is located about 1800 m southeast of the center point. It can be seen that within the study area, the She ethnic group tends to be more distributed in the northeastern region and the Han ethnic group tends to be more distributed in the southwestern region, and a certain degree of spatial differentiation still occurs between the two in the process of mixing. In terms of the size of the distribution range, the standard deviation ellipse area of She settlements is 160.3 km^2^, while that of Han settlements is 334.5 km^2^. Therefore, the distribution range of She settlements is smaller than that of Han, which is also consistent with the results of the previous analysis [45]. In terms of distribution direction, the differences between the two are not significant. The turning angle (θ) of the standard deviation ellipse was 90.0° for the She settlement and 83.1° for the Han settlement, which was also related to the overall topographic distribution of the study area with a west-east orientation.

### 4.4. Kernel Density Estimation Results

The kernel density analysis of all settlements showed (Figure 7) that the maximum density of settlements in the study area was 2.027 settlements/km^2^, which was distributed in the Tianzhushan area. The areas with lower density of settlements (0~0.178 settlements/km^2^) are mainly located in the mountainous areas with higher altitude, which shows that the distribution of human settlements is mainly influenced by the topography and water sources. The analysis of the nuclear density of She settlements in the study area clearly identifies two main gathering areas of the She, namely the northern area of Zhanping Township and the northern area of Taiyuan Township as the core of the main gathering place of the She, with a density of 1.067 settlements/km^2^. This area is likely to be a link between the She of Zhanping Township and Taiyuan Township, and belongs to the product of settlement spreading. According to the field survey of the area by related scholars, many She people in Taotian village in the westernmost part of Zhanping Township came from Lingshan County over the mountain range, and the estimated nuclear density of She settlements could also indicate this connection [46]. The kernel density analysis of Han settlements further argues for the important influence of water sources on the distribution of settlements, as seen in Figure 7c, where most Han settlements are distributed in strips along rivers.

## 5. Discussion

### 5.1. Influence Factors and Distribution of Settlements

From the perspective of the province where the whole research area is located, the spatial distribution of rural settlements is mainly affected by natural and economic conditions. Among them, rural settlements in Jiangxi Province are mainly distributed in plain areas. It is obvious that there are better farmland and transportation conditions in plain areas, which will promote the distribution of rural settlements. As the She ethnic minority inhabited areas in the study area are mainly distributed in mountainous areas, the overall village distribution density in this area is smaller than that in Poyang Lake Plain [47]. There are also many ethnic minorities in western China. The distribution of villages among different ethnic groups in these regions is also different. For example, the research on the distribution differences of Tibetan, Tu and other ethnic settlements in the Hehuang Valley area of the Qinghai-Tibet Plateau shows that Tu ethnic settlements will choose to be located in the humid areas at low altitude, while Tibetan villages will be located in the areas at relatively high altitude [6]. The main reason for this phenomenon is that there are differences in the production modes of different ethnic groups. The Tu people are more engaged in agricultural production, so they choose the humid areas with relatively low altitude. Tibetans are mainly engaged in animal husbandry, so the altitude of the distribution area is relatively high. According to the results of this study, the spatial distribution of She people in southeast China is also due to the differences in production methods. Before the 1950s, many She ethnic groups in the study area were engaged in hunting or semi-agricultural and semi-hunting production. It can be seen that the mode of production is related to the spatial distribution of residential areas. The distribution of residential areas is affected by various factors. In addition to the natural conditions themselves, the convenience of production activities is an important factor affecting the spatial distribution of residential areas.

### 5.2. Physical Geography and Settlements

Natural physical geographical factors have a direct impact on the spatial distribution of traditional villages, especially in the traditional period of underdeveloped technology. Among the main environmental factors affecting the spatial distribution of villages, altitude, rivers, and topographic features are the main factors.

#### 5.2.1. Elevation Distribution Characteristics

The results of GIS elevation analysis show (Appendix A) that the She and Han settlements show different distribution patterns with increasing elevation. The plain areas below 200 m are all Han settlements, 15 in total; the hilly areas between 200 and 500 m have 39 She settlements and 67 Han settlements; the low mountain areas between 500 and 800 m have 27 She settlements and 68 Han settlements; the areas between 800 and 1200 m are all Han settlements, 15 in total; the areas above 1500 m is only one Han settlement. It can be seen that the plain areas in the study area are Han distribution areas; the hilly areas and low mountain areas are mixed She and Han areas, but the number of She settlements exceeds that of Han.

Overall, the mean elevation of Han settlements is 403 m and that of She settlements is 473 m, indicating that the Han distribution area is lower than that of the She. However, it is worth noting that the 16 Han settlements distributed in the higher altitude >800 m mountainous areas are settlements formed due to the establishment of state forestry sites after liberation, and are not settlements formed in a completely natural process. And there are no settlements in areas >1250 m altitude, which also indicates that both She and Han are restricted by terrain and topographic elements in the process of agricultural development and land resource utilization [45]. As the altitude rises, all kinds of agricultural production will also be affected, so there will be more settlements distributed at lower altitudes and in plain terrain conditions.

#### 5.2.2. Distribution Characteristics along the River

The results of the buffer zone analysis with the river polyline file in the study area showed that there were 89 Han settlements within 800 m from the water source, accounting for 79% of the total, while there were only 23 She settlements, accounting for 21% of the total. Considering the convenience of water use for general residents, it is not very convenient to use the river water source at a distance of >200 m from the river. Within the nearest 200 m from the water source, there are 30 Han settlements, accounting for 81%, and 7 She settlements, accounting for 19%. It can be seen that there are obvious differences between She and Han settlements in terms of water source differentiation, and Han settlements are closer to water sources (Appendix A).

#### 5.2.3. Topographic Features Indicated by Place Names

Different geomorphological forms can affect the naming of habitation points, and many studies have shown that different toponymic elements can indicate different geomorphological and topographic features. In this study, we create a topographic profile centered on the place name (Appendix A) by using DEM data and spatial information of various place names in the study area to illustrate the topographic indication of the place name. The results show that the places named with “*keng* (pit)” show the characteristics of “high around and low in the middle” in the profile; the profile of the places named with “*ping* (flat)” basically belongs to gentle slope, which is consistent with the meaning of ping. In the vicinity of the sites named after “*shan* (mountain)” and “*ling* (ridge)”, there are basically mountain ranges.

### 5.3. Environmental Adaptation and Sustainable Development Implications

Sustainable development includes many aspects, such as socio-economic factors, human changes and adaptation to the environment, and so on. This study focuses on the evolution of man-land relationship in a long time scale. A study of the cultural landscape of a typical gathering area of the She people in southeastern China reveals that the geographical names of this region have more typical characteristics of adaptation to the geographical environment. The village names have obvious naming patterns in terms of topography, flora and fauna, and the surnames of the population in their settings. The place names in the study area are named with elements directly related to the local natural environment, which can reflect the survival of local residents in the mountainous environment. Since the study area is located in the hilly mountainous region of southeast China, there are few place names that directly reflect “water”, but the names of “orientation” and “*yang* (sun)” are particularly important in the mountainous region. Cultural landscapes are more common.

The spatial distribution pattern of the settlements in the study area has a certain pattern, and the names of “orientation” (Figure 3b) and “family name” (Figure 3a) are distributed throughout the study area, which is generally consistent with the overall distribution trend of the toponymic points. The names named after “plant” elements are mainly distributed in the plains and low elevation areas, which probably indicates that these places are more suitable for agricultural production and human activities. The elevation of the settlement distribution has obvious characteristics of ethnic group differentiation, with the mean elevation of Han settlement being 403 m and that of She settlement being 473 m, indicating that the Han distribution area is lower than that of the She. Among them, 57.4% of the total number of the topographic points are distributed between 300 m~600 m above sea level, which is also related to the historical upstream farming and hunting livelihood form of the She people mainly relying on hilly mountainous areas [45]. In contrast, the areas near rivers with relatively low terrain are dominated by Han settlements. The results of the spatial pattern analysis also show that Han and She have a spatially east-west divergence, i.e., the center of gravity of Han distribution is in the western part of the study area and the center of gravity of She distribution is in the eastern part of the study area. The result of this phenomenon is probably due to the fact that historically the She ethnic group migrated gradually from the mountainous areas of Fujian in the east to the Jiangxi region in the west, which eventually led to the current situation of mixed ethnic groups.

## 6. Conclusions

GIS spatial analysis and spatial statistics show that in the evolution of regional human-land relations, different degrees of ethnic development have a more obvious influence on the spatial distribution pattern of settlements. And the names of different settlements are influenced by the natural environment, reflecting a more obvious spatial heterogeneity. The main conclusions of this study are as follows.

(1) Han settlements and She settlements have obvious spatial differentiation, and in general the Han distribution area is lower than that of the She. Han settlements are mainly distributed in plain areas along rivers with elevations less than 200 m; She settlements are mainly distributed in hilly areas (200~500 m) and low mountain areas (500~800 m). Among them, the area of 200~800 m above sea level is a mixed area of She and Han, and the Han settlement is closer to the water source than the She settlement. The Han settlements are mainly distributed along the rivers in strips, while the She settlements are mainly distributed in areas farther from the rivers.

(2) Regional place names have more typical characteristics of adapting to the geographical environment. The names of villages have obvious naming patterns in terms of topography and geomorphology, flora and fauna, and the surnames of the population in their settings. In the spatial distribution pattern of settlements, the results of sample analysis and nearest neighbor index analysis show that both Han and She settlements are clustered in the spatial distribution pattern, and the distribution of She settlements is more clustered than Han, and there are more dense settlements in a certain spatial scale.

(3) The nuclear density estimates show that the She have 2 main areas of concentration, namely the northern area of Zhanping Township and the northern area of Taiyuan Township. Outside the two concentration areas there are some scattered distribution areas, a phenomenon that is likely the result of population dispersal. These She agglomerations have a more consistent geographic environment, with higher elevation and belonging to the mountainous hilly area. This also reflects that with similar geographic environment areas, the population may have convergence in migration.

(4) Regional cultural landscape is the result of the development and evolution of human-land relationship, and the comprehensive analysis of cultural landscape can understand the process of human-land relationship in a small region to a certain extent. Regional settlements have certain geographical environment indicative in naming, spatial pattern, elevation divergence, and relationship with rivers, which can reflect the environmental adaptation process of human activities.

## Figures and Tables

**Figure 1 ijerph-20-02737-f001:**
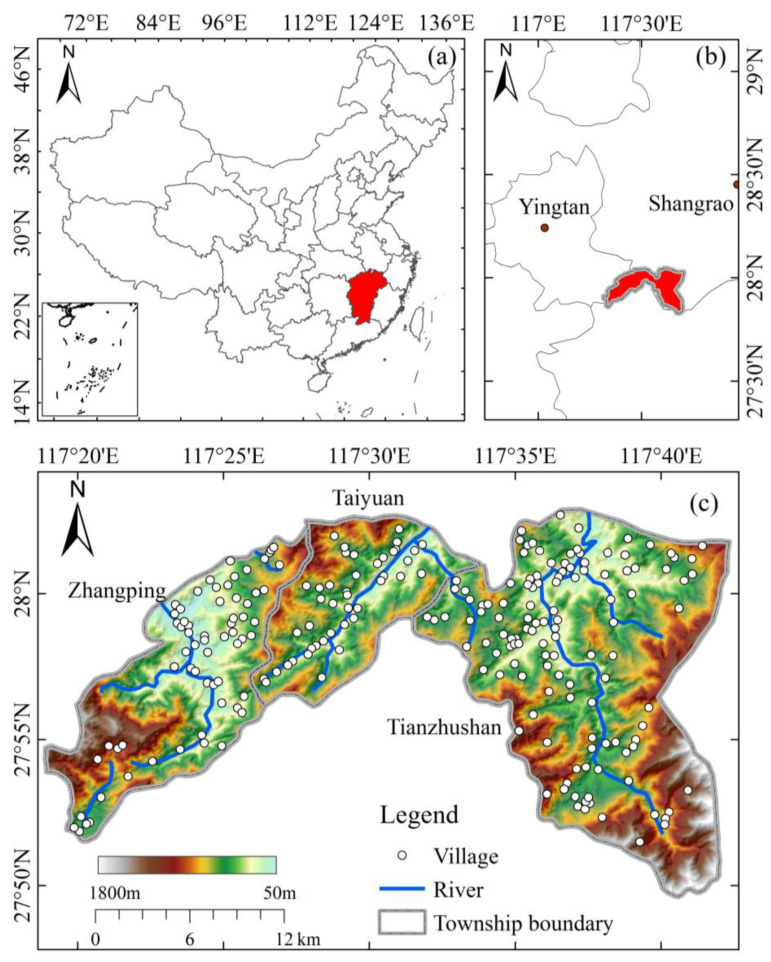
Topographical map of the study area. (**a**) Location of Jiangxi Province in China; (**b**) The location of the research area in Jiangxi Province; (**c**) Spatial distribution of village.

**Figure 2 ijerph-20-02737-f002:**
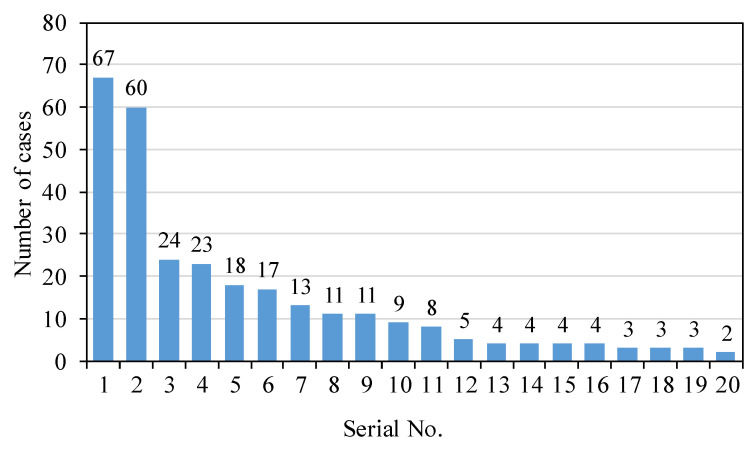
Frequency map of toponymic elements (Serial No. 1–20 represent family name, orientation, *keng* (pit), plant, event, animal, *ping* (flat), *shan* (mountain), *wu* (dock), *ling* (ridge), *yan* (rock), *yang* (sun), *pai* (row), *fan* (farmland), *pong* (clump), *tan* (beach), *gang* (hillock), *tian* (paddy field), *ling* (ridge), *ao* (depression), respectively).

**Figure 3 ijerph-20-02737-f003:**
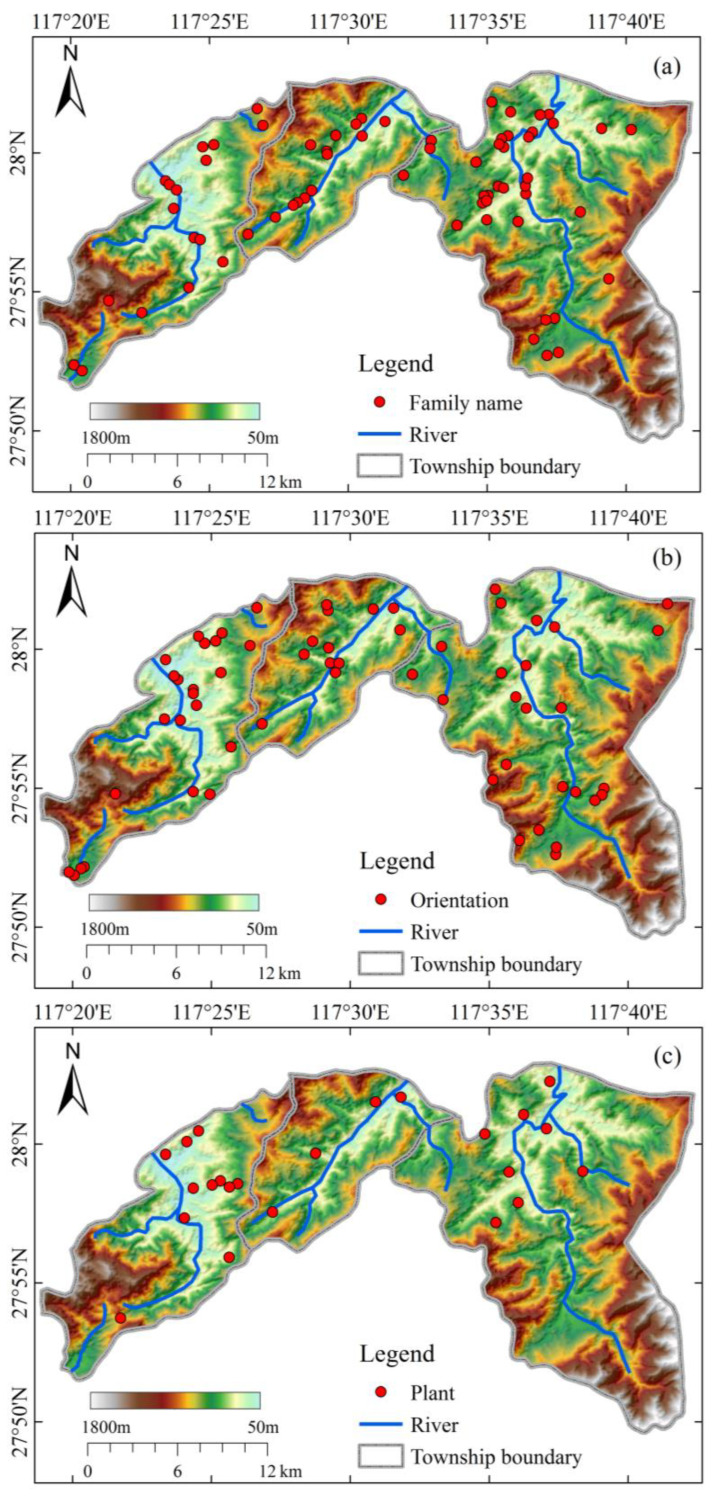
Distribution status of different types of toponymic points ((**a**), family name; (**b**), orientation; (**c**), plant).

**Figure 4 ijerph-20-02737-f004:**
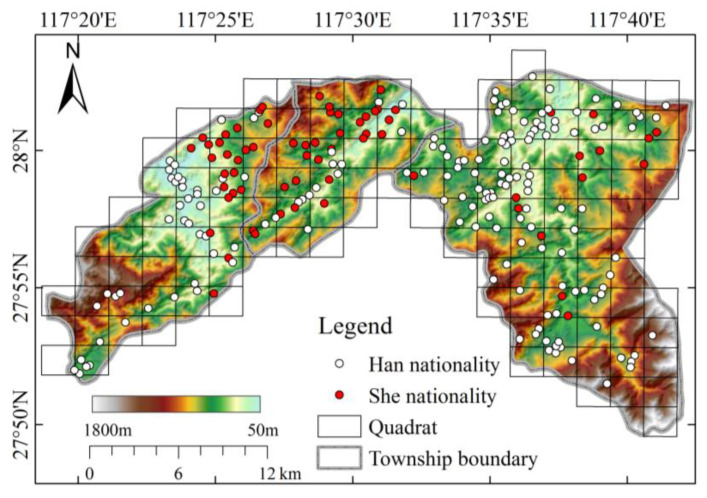
Quadrat distribution in study area.

**Figure 5 ijerph-20-02737-f005:**
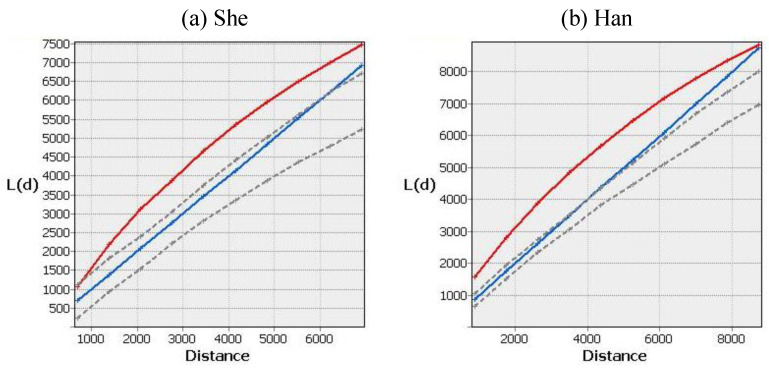
Scale features of spatial distribution of She nationality (**a**) and Han nationality (**b**) settlements. (Red line is the value of the L(d) function, blue line is the diagonal line, and gray line is the confidence interval.).

**Figure 6 ijerph-20-02737-f006:**
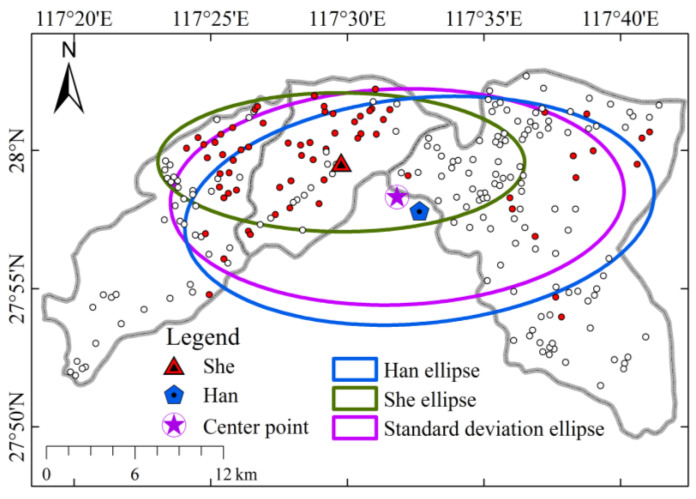
Standard deviational ellipse map of human settlements.

**Figure 7 ijerph-20-02737-f007:**
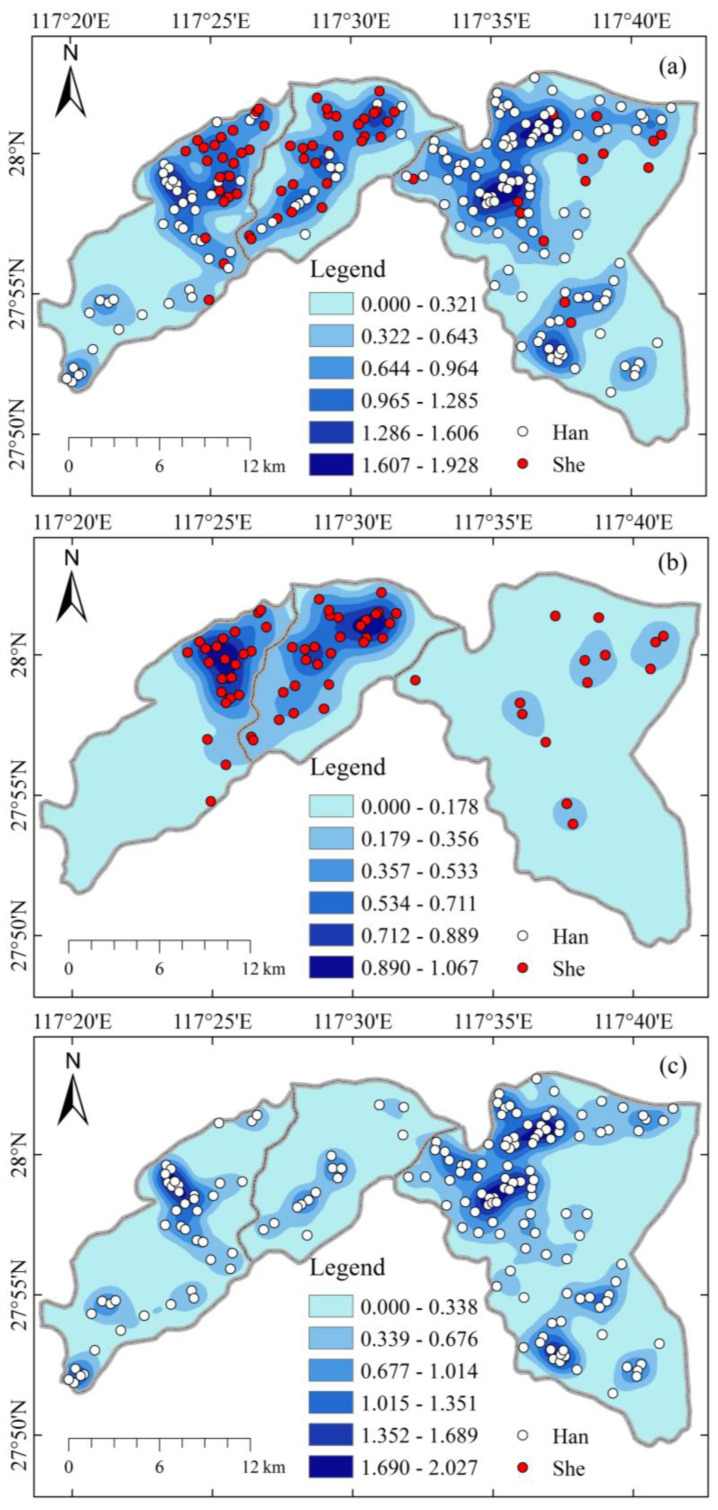
Kernel density estimation map of human settlements. (**a**) The spatial distribution of all settlements; (**b**) the spatial distribution of She settlements; (**c**) the spatial distribution of Han settlements.

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
