# Peer review of "Environmental Adaptation in the Process of Human-Land Relationship in Southeast China’s Ethnic Minority Areas and Its Significance on Sustainable Development"

_ijerph, 2023, doi:10.3390/ijerph20032737_

Round 1

Reviewer 1 Report

This study is based on GIS and spatio-temporal statistical techniques, combined with the analysis of toponymic cultural landscapes, to study ethnic minority regions of southeastern China. This manuscript is well organized and the drawn conclusions are coherent with the obtained results. The references should be updated to include more recent studies. The discussion should be completely rewritten. You must discuss your results with other studies already published. It is incredible that in a discussion there are not one references. Please, rewrite well the discussion!

Lines 33 - 34: To arrange the keywords alphabetically.

Lines 105 – 124: Please, describe better the hypothesis and predictions of your study.

Lines 106 – 107: I think that you should add this important and recent reference as example to support your sentence: “scholars have begun to use quantitative analysis methods to study the evolution process of man-land relations at different scale”. I would like to suggest:

Fraissinet, M., et al., (2022). Responses of avian assemblages to spatiotemporal landscape dynamics in urban ecosystems. Landscape Ecology, https://doi.org/10.1007/s10980-022-01550-5.

Lines 225 – 227: I think that you should add this important and recent reference as example to support your sentence: “The basic principle is to estimate the theoretical distribution of sample points in a region by means of a kernel density function, and to convert the discrete sample point density into a density value that is continuously distributed in space.”. I would like to suggest:

Bosso, L., et al. (2022). The rise and fall of an alien: Why the successful colonizer Littorina saxatilis failed to invade the Mediterranean Sea. Biological Invasions, 24(10), 3169-3187.

Figures 8,9,10: Please, move them in the supplementary materials.

Lines 343 – 416: The discussion should be completely rewritten. You must discuss your results with other studies already published. It is incredible that in a discussion there are not one references. Please, rewrite well the discussion!

Reviewer 2 Report

The authors studied the human-land relationship reflected by Han and She ethnic settlements in a typical region in southeast China, and tried to understand the spatial distribution patterns of Han and She and further the environmental adaptation of human activity in ethnic minority areas. They analyzed the place names of these settlements related to features of topography, plants, and surnames of population to explain relationship between human culture and landscape. They also compared the elevation, distance from river of Han and She settlements, and used several GIS methods to evaluate the spatial distribution of them. They concluded that the regional settlements and landscape could reflect the environmental adaptation process of human activities. The topic comparing Han and She settlements that are adapt to geographic environment is interesting, and the methods are detailed. However, I have some questions about the manuscript.

1.     There should be more background information about Han and She mixture in the study region, for instance, when and how Han and She mixed, which one was settled in this region earlier, and whether Han mixed into She or She mixed into Han. In addition, are there any ethnical customs of She people that could affect their settlements such as higher in elevation and farther from waters (as well as Han)? And it could be better to understand the meaning of this study if the specifical cultural relationship between Han and She in the study regions are pointed out. I noticed that the section “2. Overview of the study area” have some information, but it seemed not enough. Whether it could be better to add this part into section “Introduction”?

2.     The “Introduction” section should be more structured, and some information not closely related to the topic should be simplified, for example, lines 46-54, lines 73-85. Previous studies about other ethnic minorities (lines 96-101) were essential, but the effects of topography on ethnic minorities were not clearly mentioned. It is also lack of previous studies about She, which will help to understand the interaction between landscape and She and Han settlements.

3.     Both the title and the “5.4” section have mentioned sustainable development implications, but the economic-social data are not analyzed and discussed in the manuscript. Almost all the data analyzed in section “Methods” were geographical data. If some economic-social data, agricultural, human utilization data were added, it will be more reasonable to support the significance on sustainable development.

4.     The discussion part seems have lots of results. Is it possible to move these results to “Results” section, and make “Discussion” section more structure?

Other minor comments are as follows:

Line 88-91: Replicate to line 82-84

Line 101-104: Why ethnic dwellings in large areas is due to special historical processes, but that in ethnic minority areas is due to man-land relationship?

Line 108-110: Please add the references.

Line 113-114: As mentioned before, it better to add related studies about She.

Line 131: Replicate to line 114.

Line 136: Why the two places are first established as She townships? If the reasons are highlighted, the importance to select the sampling regions are more understandable.

Line 157: Is there any standard of the map of Jiangxi Province? If available, it should also be cited here.

Line 242: Whether the place name “Xinwuli” is a type error? Because “Xin” is not meaning “West”, but “Xi” is.

Line 244-245: Whether “60 times” could be written as “n=60”? The same as follows.

Line 250, 274, 286, 295, 377: The methods from ArcGIS platform had been described in “Methods” section. I think in these lines there’s no need to describe them again.

Line 317-318: Please add the references.

Line 335-338: Please add the references.

Line 354-355: Please check whether Han distribution area (elevation 503 m) is lower than She (elevation 473 m)? Is here a mistake? Because it is not consistent with following results (line 404-405)

Line 360: There is no agricultural data to support the indication.

Line 369-370: Why did not give results in 400m, 600m?

Line 374: In Figure 10, there is only one place name for each topographic landscape. Why did not analyze and show all the place names according to their topographic landscape? (for example, whether all place names with “Keng” are accord with Figure 10a? whether all place names with “Ping” are accord with Figure 10b? and so on)

Line 398-400: It better to mention the related result figures.

Line 406-409: Please add the references.

Line 410: Han or Han Chinese?

Round 2

Reviewer 1 Report

Well done!

Reviewer 2 Report

The authors have considered my comments and revised the MS well.